# Efficacy of Individualized Sensory-Based mHealth Interventions to Improve Distress Coping in Healthcare Professionals: A Multi-Arm Parallel-Group Randomized Controlled Trial

**DOI:** 10.3390/s23042322

**Published:** 2023-02-19

**Authors:** Hannes Baumann, Luis Heuel, Laura Louise Bischoff, Bettina Wollesen

**Affiliations:** 1Department of Biopsychology and Neuroergonomics, Technical University Berlin, 10623 Berlin, Germany; 2Institute of Human Movement Science, University of Hamburg, 20146 Hamburg, Germany; 3Medical School Hamburg, Institute of Interdisciplinary Exercise Science and Sports Medicine, 20457 Hamburg, Germany

**Keywords:** biofeedback, tailoring, digital health, nurses, ECG, sensor, eHealth, heart rate variability, accelerometry, health app

## Abstract

Detrimental effects of chronic stress on healthcare professionals have been well-established, but the implementation and evaluation of effective interventions aimed at improving distress coping remains inadequate. Individualized mHealth interventions incorporating sensor feedback have been proposed as a promising approach. This study aimed to investigate the impact of individualized, sensor-based mHealth interventions focusing on stress and physical activity on distress coping in healthcare professionals. The study utilized a multi-arm, parallel group randomized controlled trial design, comparing five intervention groups (three variations of web-based training and two variations of an app training) that represented varying levels of individualization to a control group. Both self-reported questionnaire data (collected using Limesurvey) as well as electrocardiography and accelerometry-based sensory data (collected using Mesana Sensor) were assessed at baseline and post-intervention (after eight weeks). Of the 995 eligible participants, 170 (26%) completed the post-intervention measurement (Group 1: N = 21; Group 2: N = 23; Group 3: N = 7; Group 4: N = 34; Group 5: N = 16; Control Group: N = 69). MANOVA results indicated small to moderate time-by-group interaction effects for physical activity-related outcomes, including moderate to vigorous physical activity (F(1,5) = 5.8, *p* = ≤0.001, η^2^p = 0.057) and inactivity disruption (F(1,5) = 11.2, *p* = <0.001, η^2^p = 0.100), in the app-based intervention groups, but not for step counts and inactivity. No changes were observed in stress-related heart rate variability parameters over time. Despite a high dropout rate and a complex study design, the individualized interventions showed initial positive effects on physical activity. However, no significant changes in stress-related outcomes were observed, suggesting that the intervention duration was insufficient to induce physiological adaptations that would result in improved distress coping.

## 1. Introduction

Occupational psychosocial stress can increase the risk of developing psychological, musculoskeletal, or cardiovascular diseases [1,2]. Healthcare professionals are particularly prone to experience exceptionally high levels of occupational stress [3] leading to serious individual, organizational, and societal problems as for example shortages of skilled professionals [4]. In one study by Couarraze and colleagues, for instance, healthcare professionals indicated stress levels 25.8% higher than the general population [5]. Additionally, more than one in four nurses consider leaving the profession [6]. Additionally, healthcare institutions often fail to retain long term personnel, exposing healthcare professionals to a vicious cycle of stress [7]. This is consistent with findings from previous studies indicating higher occupational distress in health care occupations than in other occupations [8]. Recurring stressors in health organizations include high work demands, leadership style, few participation opportunities for work structuring, emotional burdens, lack of appreciation, and work–family conflicts [9,10]. In turn, these stressors may influence recreational activities of the affected persons. For instance, sleep and physical activity levels have been found to be poor in stressed individuals [11,12]. Frequent or chronic occupational distress results in serious health consequences. If work-related demands outweigh individual, social, and organizational resources [13] affected individuals may incur psychological and physiological consequences such as sleep disorders, gastrointestinal complaints, burnout, diabetes, and coronary heart disease [14,15,16,17]. In severe cases, inability to work can lead to long-term sickness absenteeism [18].

In general, psychosocial stressful stimuli activate neuronal, neuroendocrine, and endocrine pathways [19]. Thus, a physiological response to stress occurs, among others, at the neurological level, through receptors of the sympathetic nervous system that stimulate the sympatico-adrenomedullary axis. The hormones adrenaline and noradrenaline are released in the adrenocortical medulla, leading to an increase in heart rate and a decrease in heart rate variability (HRV) under physical or psychological stress [20]. Such biological responses to stressful stimuli may be adaptative. However, extreme, frequent, or chronic activations of stress axes may be detrimental to health and may be assessable via HRV [21,22]. Chronically low HRV is associated with impaired regulatory and homeostatic functions of the autonomic nervous system, which reduce the body’s ability to cope with internal and external stressors. For instance, individuals with lower HRV were more likely to report poorer quality of sleep in the context of chronic stressor exposure than individuals with higher HRV [23]. Thus, HRV measurement is a noninvasive method that can be used to measure the autonomic nervous system in a variety of settings [24]. For instance, it has been shown that in response to stress-inducing tasks, such as the Trier Social Stress Test, individuals show low parasympathetic activity, characterized by a decrease in High Frequency Power (HF) and an increase in Low Frequency Power (LF) HRV values [24,25,26]. The Standard Deviation of Normal-to Normal heart beats (SDNN value) represents an index of physiological resilience to stress. When HRV is elevated and irregular, SDNN increases. On the other hand, chronic (occupational) stress is linked to a decrease in the SDNN [24,27]. A low Root Mean Square of Successive Differences (RMSSD) value can also be an indicator of stress, with values lower in chronically stressed vs. non-stressed individuals [20,26,28]. However, it should be noted when evaluating HRV data that—beyond psychological stress—certain influencing variables must be considered. Age has a significant influence on HRV. It initially increases with age, peaks in young adults, and then decreases with increasing age [29,30]. In addition, BMI correlates positively with sympathetic activity [31] and thus negatively with HRV [32,33], whereas regular physical activity is associated with an increase in HRV [34].

While health complaints are frequently observed, there are personal and organizational resources which can improve resilience toward occupational stress [35,36]. Personal resources concerning coping qualities include social support, coping style, self-efficacy, and optimism [37]. Pertaining to organizational resources, a recent systematic review identified supervisor support, job autonomy, and provision of work equipment to minimize stress [38]. Stress may also affect healthcare professionals differentially. For instance, nurses exhibit fewer health behaviors [e.g., physical activity] than physicians, pharmacists, and administrative health personnel [39]. According to Gerber and Pühse [40] physical activity may exert a stress-buffering effect and thus protect against physical and psychological illness. Although a variety of stressors exist in healthcare professionals, evidence suggests that perceived stress can be reduced through the participation in stress management interventions. For instance, mindfulness programs improve quality of life, anxiety, stress perception, and sleep quality [41,42]. Physical activity-based studies showed improvements in autonomous nervous system function [43] and accelerometric factors such as steps per day [44], BMI, sedentary behavior, MET, and physical activity levels [45]. Physical activity can help reduce stress by improving physical and emotional well-being across at least three pathways. (1) It reduces the release of stress hormones: during physical activity, endorphins are released, which increase well-being and decrease the release of stress hormones such as cortisol. (2) Physical activity promotes relaxation: physical activity can help relax muscles and release tension that builds up during stress. (3) Physical activity improves mental health: physical activity can help reduce symptoms of depression and anxiety and improve self-esteem [46,47]. Other than physical activity, the effectiveness of breathing exercises such as diaphragmatic breathing [48], progressive muscle relaxation [49], meditation [50], yoga [51], gratitude journaling [52], listening to music [53], and autogenic training [54] is evident.

Within the health care sector, efficacious stress reduction programs include yoga and qigong [47], cognitive-behavioral interventions such as resilience training [55], mindfulness-based stress reduction (MBSR) [56], or multimodal combinations of aforementioned intervention types [57]. According to the literature, four tasks that need to be completed when designing individual-level interventions for healthcare professionals are identifying barriers, selecting intervention components, using theory, and engaging end-users [58]. The length of a health intervention will depend on a variety of factors, including the specific health problem being addressed, the goals of the intervention, and the resources available. Some health interventions may be short-term, lasting only a few days or weeks, while others may be long-term, lasting months or years [59]. In general, it is important to carefully consider the length of a health intervention and to ensure that it is sufficient to achieve the desired goals. Short-term interventions may be appropriate for addressing acute health problems or for providing targeted support for specific populations. However, long-term interventions may be necessary for addressing chronic health problems or for addressing more complex health issues that require sustained support and intervention [58]. It is also important to consider the sustainability of a health intervention and to ensure that it can be maintained over the long term. This may involve developing strategies for funding, staffing, and resource management, as well as engaging community members and other stakeholders in the planning and implementation of the intervention.

Despite the plethora of studies confirming the efficacy of stress reduction interventions, the evidence for health personnel is insufficient. Study rigor issues, for instance low total intervention time, small sample sizes, and high dropout rates, undermine intervention quality [60]. Further, elevated risk of bias due to lack of both appropriate study designs and follow-up measurement points are common [47]. The poor evidence base in the field of study is due to organizational, social, and individual reasons. According to Zhang et al. [61], participation in health promotion campaigns in health care facilities is often aggravated by various barriers. Specifically, poor communication between management and staff, colleague peer pressure, insufficient staffing, top-down decision-making, and budget constraints can impede participation rates. Additionally, healthcare personnel are difficult to reach due to low motivation to change, low self-efficacy, and high psychological and physiological demands [62]. Moreover, due to differences in individual and organizational resources, stress management interventions should be individualized to the specific needs of participants. One possibility is to categorize subjects in terms of coping style when facing challenging work situations [63]. Further, individual preferences for health promotion are apparent. For instance, health and other non-health related outcomes (e.g., the value of a healthy future self and time costs, respectively) have differential impacts on the decision to engage in stress management [64]. Thus, one-size-fits-all interventions [65] should not be adapted for vulnerable populations as intervention success is limited [66]. In sum, to counteract stress effects in health personnel, low-cost, easy-to-implement, setting-specific, and need-individualized health promotion interventions are necessary. One way to address these issues are digital interventions, especially when delivered via a mobile device (mHealth). Recent developments and studies highlight the opportunities of digital interventions to address the described concerns for implementing and evaluating interventions in the health care sector and the current stage of change readiness. mHealth interventions yield the potential to address stress in a low-cost, easy-to-implement fashion [67] with existing evidence for stress-reducing effects in different occupational settings [68]. Interestingly, internet-based interventions have been rarely implemented in the healthcare sector so far [69]. One systematic review by Kim et al. (2020) found that mHealth interventions were effective in reducing stress and improving mental health outcomes, such as anxiety and depression, among healthcare professionals. Another study by Kim and colleagues revealed that a mobile app intervention was effective in reducing burnout symptoms and improving job satisfaction among nursing staff. Similarly, a randomized controlled trial by Kang and colleagues [70] found that a mobile app intervention was effective in reducing stress and improving mental health outcomes, such as anxiety and depression, among medical residents. These studies suggest that mHealth interventions have the potential to improve mental health outcomes and reduce stress among healthcare professionals.

Digital health promotion programs can come in different modalities: web-based trainings (WBT) are presented on a secured online platform and assessed through an internet browser either on a smartphone or on a computer/laptop [71,72], whereas app-based interventions are delivered via smartphone application only [69]. However, there are also hybrid forms such as web apps. One example for a hybrid approach could be the mCARE project by Rabbani and colleagues [73]. The project discusses a data-driven validation of a mobile-based care project (mCARE) aimed at helping children with Autism Spectrum Disorder (ASD) in Low and Middle-Income Countries (LMICs). The results showed that the mCARE project had a positive impact on the children’s symptoms and behavior and was effective in reducing the burden on caregivers. The study provides evidence for the potential of mobile technology to improve access to care for children with ASD in LMICs. mHealth interventions can be a low-threshold opportunity for health promotion and are a promising possibility to achieve prevention goals [74]. The free allocation of time and flexibility of availability were evaluated on a positive note. Combining such apps with so-called “wearables”, such as smartwatches or fitness trackers, could be a promising approach to continuously record health data and thus constitute various opportunities in the context of prevention work (gamification, just in time adaptive interventions). By implementing wearable devices into mHealth applications (apps), health-related data (e.g., sleep patterns, eating patterns, and exercise) could be recorded, and respective need-tailored interventions be derived [75]. Previous studies already showed positive effects of stress apps on wellbeing. Harrer et al. [76] for example found that app-based stress management interventions improved stress, anxiety, and depression in college students. Another example stems from research by Economides et al. [77] who found that a mindfulness app intervention reduced stress and irritability, while it also increased positive affect. At the same time, expectations towards health apps are high; 70% of health app users believe that these can strengthen self-motivation, and 56% think that app use can improve health education [78]. In order to establish long-term health behavior changes in healthcare workers, an elevated level of adherence motivation during the intervention implementation is necessary, and therefore individualized approaches may be beneficial. Often, the adherence to digital health promotion programs is low, which reduces their effectiveness [79]. Individual tailoring [80,81] or gamification could be approaches to address this problem. A meta-analysis showed that web-based individualized interventions clearly outplayed generic interventions with respect to health behavior change [81]. In particular, non-individualized interventions were found to decrease user satisfaction [82]. However, the definition of an individualized health app remains unknown due to the lack of a framework for individualized app elements [83]. In the context of mHealth, individualization is defined as an adaptation to the needs or special circumstances of an individual and the lack of such is cited as one of the main barriers that prevent patients from behavior change [84,85]. Individualized interventions (sometimes also called adaptive, needs-specific, target group-specific, tailored, or personalized interventions) offer a potential way of delivering person-centered interventions by varying levels of individual needs and empowering individuals to monitor their health actively [86]. In one of the few reviews that address this issue, the authors enumerated the individualized elements in the app and determined the level of individualization of mHealth intervention [87]. The evidence of this review is clear; however, there is a wide range of potential approaches for individualization, and these are often accompanied by established behavior change mechanisms.

Potential opportunities for individualization are (1) the adaptation of intervention content to individual needs for behavior change, (2) individual coaching based on intervention results, (3) direct biofeedback via app and sensor interfaces, (4) visualization of health data, e.g., in the form of health dashboards or health reports, or (5) the adaptation of content based on psychological characteristics, such as personality traits. Needs assessment, health reports, and coaching have already been discussed above. The most commonly used devices in biofeedback differed depending on the outcome of the evaluation. For physical activity parameters ActiGraph accelerometer (Actigraph, LLC, Pensacola, FL, USA, SenseWear wristbands (BodyMedia, Inc., Pittsburgh, PA, USA), Actical (Mini Mitter Co., Inc., Bend, OR, USA), or Active style Pro (Omron Healthcare Co., Ltd., Kyoto, Japan) are the most common [88]. Accelerometer sensors are devices that measure acceleration or changes in movement or position. They are commonly used in a variety of applications, including smartphones, fitness trackers, and wearable devices.

Advantages of accelerometer sensors include: (1) High sensitivity: Accelerometer sensors are highly sensitive and can accurately measure even small movements or changes in position; (2) Compact size: Accelerometer sensors are small and lightweight, making them easy to incorporate into a variety of devices and systems; (3) Low power consumption: Accelerometer sensors have low power requirements and can operate for long periods of time without needing to be recharged; (4) Versatility: Accelerometer sensors can be used in a wide range of applications, including motion sensing, activity tracking, and gesture recognition. Potential disadvantages of accelerometer sensors include: (1) Limited accuracy: While accelerometer sensors are highly sensitive, they may not be as accurate as other types of sensors, such as gyroscopes, in certain applications; (2) Vulnerability to noise: Accelerometer sensors may be prone to interference or “noise” from external sources, which can affect their accuracy and performance; (3) Limited range: Accelerometer sensors may have a limited range of movement or acceleration that they can measure, depending on the specific device or application. Nevertheless, accelerometer sensors are useful and versatile devices that have a wide range of applications. However, it is important to consider the potential limitations and challenges of using accelerometer sensors in order to ensure the best possible performance and accuracy.

For stress-related parameters, photoplethysmography (PPG)-based wearable devices such as earlobe sensors, blood pressure monitors, finger bracelets and wristwatches or electrocardiogram (ECG)-based devices such as chest belts or patches are commonly used, with the latter exhibiting higher sensitivity and specificity values [89]. ECG sensors are devices that measure the electrical activity of the heart and are used to diagnose a variety of cardiac conditions. Advantages of ECG sensors include: (1) Non-invasive: ECG sensors are non-invasive and do not require any penetration of the skin or tissue, making them relatively safe and painless to use; (2) High sensitivity: ECG sensors are highly sensitive and can accurately measure the electrical activity of the heart, even in the presence of noise or interference; (3) Portability: ECG sensors are portable and can be used in a variety of settings, including hospitals, clinics, and home care settings; and (4) Versatility: ECG sensors can be used to diagnose a wide range of cardiac conditions, including arrhythmias, heart attacks, and coronary artery disease. Potential disadvantages of ECG sensors include: (1) Limited accuracy: While ECG sensors are highly sensitive, they may not be as accurate as other diagnostic tests, such as echocardiography, in certain situations; (2) Vulnerability to interference: ECG sensors may be prone to interference or “noise” from external sources, such as electrical devices or electromagnetic fields, which can affect their accuracy and performance; and (3) Limited scope: ECG sensors can only measure the electrical activity of the heart and do not provide information about the structure or function of the heart or other organs. Overall, ECG sensors are useful and valuable diagnostic tools that have a wide range of applications. However, it is important to consider the potential limitations and challenges of using ECG.

Beside sensory based biofeedback, another approach to design individualized digital solutions could be the integration of personality traits. A smartphone app that focuses on stress reduction may firstly focus on personality characteristics, as studies showed that personality characteristics are associated with specific coping behavior [90], app usage behavior, and receptivity to gamification elements [91]. Implementing personality traits into mHealth interventions offers the opportunity to systematically individualize the content. Besides the adaptation to personality, it is also necessary to address health behavior change intention. Thus, if participants are not intending to change their activity levels and stress coping behavior, the intervention might fail to succeed. However, the intervention could be tailored to target the individual at the current stage of health-related behavioral change. In summary, for the development of a digital health intervention, the specific combination of different components has to be considered. These are: (1) evidence-based feasible interventions, (2) tailoring and individualization, and (3) additional elements to gain adherence and long-term usage.

Therefore, the present study aims to compare both web-based vs. app-based and individualized vs. non-individualized stress management interventions in terms of their effectiveness. The main research question is whether eight weeks of differentially individualized sensor-based mHealth interventions (1 = WBT, 2 = WBT + Need, 3 = WBT + Need + Coaching; 4 = APP + Biofeedback, 5 = App + Biofeedback + Healthreport) focusing on stress management and physical activity can impact HRV-related stress parameters (SDNN, RMSSD, LFHF, and Baevsky Index) and accelerometry related physical activity parameters (Steps, MVPA, Inactivity, and Inactivity disruption) and therefore improve distress coping in health professions. We hypothesize that individualized interventions will have small to moderate positive effects for physical activity and stress-related outcomes in relation to distress coping in health professions, whereas non-individualized interventions will not show significant effects. To the best of our knowledge, this study complements the described existing body of research and is the first to:Scientifically validate a sensor-based mHealth intervention for distress coping in the healthcare setting.Compare the efficacy of different individualization levels of mHealth interventions by a multi-arm study design.Combine components of physical activity and relaxation techniques in an mHealth application using a multimodal intervention approach to improve distress coping among healthcare professionals.Measure multiple clinically relevant stress and physical activity-related outcomes during the intervention and adapt the content of the intervention based on these measures.Enable unrestricted implementation within the daily work routine of healthcare professionals by means of a mobile and low-threshold intervention.

## 2. Materials and Methods

### 2.1. Trial Design

This multi-arm parallel group randomized controlled trial (including five intervention groups) was conducted and described [87] according to the CONSORT guidelines [92], including the necessary extensions [93,94]. All participants of the intervention groups received a digital intervention. Both questionnaire and sensory data were assessed at baseline (T1 pre-intervention assessment) and at eight weeks (T2: Post-Intervention assessment). However, this paper only refers to the sensory data. The five intervention groups were conducted as follows (see Table 1).

### 2.2. Participants

The trial included multiple healthcare professionals (nursing staff and office workers) aged 18 years or older. No clinical patients were involved in the proposed study. An a priori power analysis with G*Power [95] indicated the necessity of at least 700 participants to show moderate effect strengths (0.25) with a beta error of 80%. The executives of collaborating hospitals, stationary elderly care facilities, and ambulatory care providers forwarded an explanatory video to their employees via in house communication networks, whereupon they voluntarily entered their contact details into an online tool to register for the study. Fluency in the German language as well as internet access via a smartphone device were prerequisites for study participation. In order to improve adherence to interventions, a user centered approach was chosen to integrate experiences and test the functionality of the app internally and externally. After agreeing to participate, numerous reminder emails were created, which were automatically sent to the participants if they failed to order the sensors or missed the registration.

To prevent selection bias, the allocation of participants to the intervention and control groups was randomized. The random allocation at individual level was conducted with the tool Research Randomizer [96] using continuous block randomization. Sets of six numbers were generated, representing the differing number of study and control groups. Each participant was then assigned the subsequent number on the block randomization list for group assignment. As participants were assigned to an intervention group or the waiting control group by lot, no further mechanisms of implementing the allocation sequence were needed. Unblinding of the data assessors was not necessary.

Trial participants were informed about which intervention group they were assigned to as they needed to receive the respective information to complete all necessary information and access the digital intervention programs. Furthermore, participants were informed in advance to ensure the intervention is implemented during working hours and outside of vacation periods. The data collection of primary outcomes was also blinded, as participants self-completed the online questionnaire, and the sensor screening was similarly conducted without the involvement of a third party, as participants self-applied the sensor to their bodies. All data analyses were conducted by blinded evaluators.

### 2.3. Interventions

There were five different intervention scenarios (study arms), each including a WBT or an app and each with various levels of individualization. The app interventions included individualization according to the AVEM personality type (work-related behavior and experience pattern) [97]. This trial became particularly complex due to the need orientation of the WBT interventions. Depending on the needs of a person, the participant was assigned to a different WBT. For example, someone with insufficient physical activity and severe obesity has been recommended a WBT for weight loss, while someone suffering from high stress levels has been recommended a WBT with autogenic training or mindfulness. For this reason, a detailed list of the content covered in the respective app or WBTs is provided in Table 2 below.

Likewise, the distinguished stress and physical activity-specific content of the interventions can be inferred from the table. The app-based study arms featured higher levels of individualization than the WBTs. The content of the app-based mHealth interventions (study arms 4 and 5) was also developed exclusively for use on a smartphone (see Figure 1a–c for insights into UI design), whereas the WBT-based mHealth interventions (study arms 1, 2, and 3) could also be accessed using a web browser on a desktop computer.

### 2.4. Outcomes

The assessment applied a selection of standardized questionnaire measures as well as sensor-based physiological and vital parameter measures (measured by Corvolution CM300 [99], which includes ECG circuit, 3-axis acceleration and rotation rate chip, air pressure chip, thoracic impedance chip, and temperature chip) [100,101]. The sensitivity of patch-based ECG sensors such as this is 93.4–97.0%, and the specificity is 95.6–98.8% [89]. Additionally, demographic characteristics, such as age, gender, and job hierarchy, were assessed via standardized questionnaires within Limesurvey 5.4.15 [computer software] [102]. The detailed description of all parameters can be found in the study protocol [87]. However, the current study focused on the sensory datasets in the stress and physical activity domains. The following parameters were considered (see Table 3).

## 3. Results

### 3.1. Flow-Chart

Based on the results of an a priori power analysis and an expected dropout rate of 20% (this appeared to be a realistic participation rate in previous intervention studies in different small- and middle-sized companies [105]), we were able to out-recruit slightly and obtain a total of 995 participants from multiple institutions. We started the eligibility assessment in June 2021 and completed the data collection in June 2022. Among the 995 eligible participants, 113 failed to respond to contact attempts, and 239 retrospectively declined to participate due to lack of time or illness.

Therefore, merely 643 participants were assigned to the study groups and received interventions (see Figure 2). Due to organizational (e.g., shift work, lack of time), technological (e.g., synchronization errors, outdated operating system), and physiological reasons (e.g., allergies, illness, or arrhythmias), a total of 258 subjects were unable to complete the baseline measurement despite receiving the sensor and the intervention. We further lost 218 subjects to follow-up measurement. This resulted in the analysis of a total of N = 170 participants, which corresponds to a total dropout rate of 74%. Study arm-wise, 16% of participants completed the post measurement after the app intervention + biofeedback, 18% after the WBT intervention, 19% after the app intervention + biofeedback + health report, 20% after the WBT intervention + need orientation + coaching, and 33% after the needs-based WBT intervention.

### 3.2. Baseline Data and Main Analysis

Table 4 reports the descriptive values and statistics of each measurement point (Pre-Intervention Assessment and Post-Intervention Assessment) and each study arm. The participants were analyzed in their original assigned groups. There were no significant differences in baseline demographics between the intervention and control groups. Furthermore, participants lost to follow-up were not significantly different from those considered. Across all study arms, there were more female than male participants in the sample. The average age of participants at baseline was 41.1 ± 10.9 years.

The group size of the study arms was not identical and varied from 34–203 participants per group at baseline to 7–69 participants per group at post assessment. The statistical analysis (MANOVA) indicated significant time*group effects for the two physical activity-related outcomes MVPA minutes as well as inactivity disruption counts.

Post hoc analysis revealed individualized app study arms (4 and 5) to be significantly different from the control group and less individualized WBT study arms (1, 2 and 3) in these outcomes. Participants in study arm 4 increased their activity time by 72.5 ± 45 min, and participants in study arm 5 by 69.3 ± 11.7 min, whereas in all other study arms and the control group, activity time decreased. A similar pattern can be seen for inactivity disruptions.

While the number of inactivity disruptions per day increased significantly in app trials by 4.9 ± 3.5 in study arm 4, and 5 ± 1.5 in study arm 5, respectively, it decreased in all other study arms and the control group. In addition, time*trialgroup (intervention vs. control group) and time*interventiontype (app vs. WBT) effects have also been tested. The results showed significant effects on the same outcome variables, although the magnitude of the effect was smaller. All stress-related outcomes (SDNN, RMSSD, LFHF, and Baevsky Index), as well as the other physical activity-related outcomes (Steps and Inactivity), did not differ significantly across measurement time points or among study arms. Figure 3 and Figure 4 visualize the significant results from Table 4. In addition to the differences in sample size and variance across variables and study arms, this illustrates the magnitude of effects.

## 4. Discussion

This multi-arm parallel randomized controlled trial aimed to investigate the efficacy of multiple differentially individualized sensory-based mHealth interventions to improve distress coping with regard to physical activity and stress related outcomes in healthcare professionals. We hypothesized that individualized interventions would have small to moderate positive effects for physical activity and stress-related outcomes in healthcare professionals, whereas non-individualized interventions would not show efficacy.

Contrary to our expectations, stress-related HRV-parameters did not show significant improvements over time, regardless of the study arm or the resulting level of individualization. Within this context, the stress buffering hypothesis assumes that physical activity and stress perception are closely related constructs [40]. However, to achieve cognitive and psychophysical adaptations through physical activity, continuous, specific training according to exercise principles is necessary for sustainable effects [106,107]. These criteria could not have been met within the guidance of the app. To gain a positive effect on HRV parameters or subjective reported stress, physical exercise such as yoga or endurance training needs to be performed on a regular basis (Bischoff et al., 2019). Aside from the challenges in obtaining the stress-related parameters (discussed in more detail in the limitations section), an 8-week intervention may retrospectively be insufficient to activate physiological mechanisms that have a stress-buffering effect. Long-term interventions may be necessary for addressing chronic stress symptoms or for addressing more complex health issues that require sustained support and intervention. An even stronger involvement of the participants would have been useful in terms of intervention mapping [59]. Moreover, the lack of supervision during the intervention in the mHealth interventions forced the participants to self-pace the intervention. This is a major disadvantage compared to supervised interventions [82,83,84]. Therefore, we conclude, that this type of mHealth intervention should include motivational aspects and guidance to do additional structured physical exercise next to the App use.

In contrast to the results on the physiological HRV based stress parameters, the interventions show positive effects on the accelerometry-based measured physical activity-related outcomes in high individualized app-based study arms (app-based digital stress management interventions with sensory biofeedback without (4) and with health report (5)). Strikingly, the small to moderate effects in physical activity typical for mHealth interventions [78] could only be shown for the outcomes of moderate to vigorous physical activity [min/day] and inactivity interruptions [counts/day] but not for those of steps [counts/day] and inactivity [min/day]. Besides the fact that the considered interventions did not have steps and inactivity reduction as a primary goal, the nature of the nursing profession could be another possible explanatory mechanism: other studies indicate higher step counts in nurses than in other occupations [108] as well as long work commutes and night shifts with long inactive periods [109]. Consequently, while a participant completes the intervention during working hours as instructed, this results in higher levels of moderate to vigorous physical activity and increased inactivity disruptions on the one hand; it inevitably results in elevated, consistent step counts due to patient work and elevated, unavoidable inactivity levels due to commutes and night shifts on the other. The findings from our study suggest that the relationship between physical activity and stress may vary depending on the context in which the activity takes place. This supports the idea of the “physical activity paradox” [110,111], which refers to the idea that the benefits of physical activity may depend on the specific circumstances in which it occurs. Our results suggest that physical activity may be perceived as more stressful when it is part of work, rather than leisure time, which suggests that interventions aimed at increasing physical activity in a work setting may not necessarily reduce stress levels. However, if physical activity is increased without also increasing stress, this could still be considered an improvement. Overall, these findings highlight the importance of considering the context in which physical activity occurs and the need to differentiate between occupational and leisure time physical activity when studying the relationship between physical activity and stress.

However, the effectiveness of an app-based intervention seems to be largely dependent on design aspects and user-centeredness. Despite all efforts to represent different levels of individualization across study arms, it could not be demonstrated which level of individualization is more effective based on effect sizes, as only both app-based interventions were able to show significant effects. With respect to our initial hypothesis, we would have assumed that study arm 1 (WBT only) failed to show effects due to lack of individualization. This idea was supported by the results. It would have been reasonable to suspect that efficacy would increase across the remaining four study arms due to increasing individualization. However, no significant effects were found for study arms 2 (need-oriented WBT) and 3 (need-oriented WBT and Coaching). Study arms 4 (biofeedback app without health report) and 5 (biofeedback app with health report) each indicated homogeneous effect sizes for the outcomes MVPA and inactivity disruption. Thus, it could be argued, that based on the results of this study, it seems to make no difference whether a health report is displayed or not. However, one possible reason for this result could also be the small sample size in the individual study arms. Due to the high dropout rate, the number of subjects was not sufficient to show the expected moderate effects according to the power analysis. The results should therefore be interpreted with caution.

Nevertheless, the findings further indicate that individualized app-based interventions with direct biofeedback and differentiation by personality structure show better efficacy than web-based trainings (WBT) accessed via the smartphone browser. However, one reason for the high dropout rate were technical complaints while using the app-based interventions. With additional effort in the technical aspects this disadvantage could be minimized. Therefore, it remains unclear to what extent the need orientation or the coaching, which were exclusive for WBT, would have resulted in a further improvement of the effect size in the app-based interventions. One possible explanation for the limited effectiveness of our intervention, in addition to the high dropout rate, is the insufficient incorporation of health behavior change strategies. While our biofeedback app included both active and passive behavior change techniques and promoted stress management skills, some of the content proposed by Bischoff et al. [112] was not implemented. Specifically, we applied individualization of app content, fulfilling common weekly goals and tasks, increasing knowledge about a healthy lifestyle, reminders for objectives, and controlling and checking progress but did not include many suggestions for activities with diaries for documentation and development of strategies or informational or instructional videos. The inclusion of these behavior change mechanisms could potentially enhance behavior change in future interventions.

### 4.1. Strengths and Limitations

To the best of our knowledge, this is the first mHealth intervention in the healthcare setting of this quality and complexity in the study design, demonstrating initial effects in the area of physical effectiveness despite a small sample size and not-to-be-despised dropout rates. Furthermore, it is the first mHealth intervention including multiple study arms with different levels of individualization demonstrating differences in efficacy. Nevertheless, the conditions of data collection were difficult, which can be seen as a possible reason for the high dropout rates. The dropout rate of 74% was almost four times higher than expected. This was not least due to the fact that during the COVID-19 pandemic it was not possible to establish personal contact with the participants. Email based communication does not seem to work well in the healthcare setting, as 113 people were excluded due to non-response. One potential contributor regarding communication issues and multiple technical inconsistencies could be the evident low level of digital literacy among nurses [113]. Although we were aware of these circumstances when designing the intervention, a potential approach for future interventions could be to provide pre-interventional training and develop digital literacy first. Other reasons for the high dropout rate could be the intervention or the measurement procedure. The participation threshold was not sufficiently low for healthcare workers. Excessive demands resulted from numerous extensive questionnaires, autonomous sensor orderings, and the proprietary installation of the app. In addition, the app did not support push notifications, and synchronization problems between the app and sensor occurred frequently. With regards to the measurement procedure, it should be noted that the intervention had a different initiation time and duration for all participants. They were instructed to wear the sensor during working hours and for at least 48 h. We were unable to identify from the data sensor wearing timing aspects and whether vacation periods were taken into account.

### 4.2. Future Research

Future interventions should use a less complex and longer-term study design to systematically demonstrate which individualization mechanisms lead to greater effectiveness of mHealth interventions in terms of distress coping. The focus of future mHealth interventions in the healthcare setting should be as low-threshold access as possible, including push notifications and ideally an on-site project coordinator who can provide technical support, establish accountability, and remind participants to follow procedures. To reduce the dropout rate, future studies could offer incentives, simplify the study process, improve face-to-face communication, monitor and address adverse events, and foster a more positive research culture.

Furthermore, machine learning elements could improve future mHealth interventions by providing personalized and data-driven solutions for health problems. Machine learning algorithms can analyze large amounts of data from various sources, such as electronic health records, wearable devices, and mobile apps, to identify patterns and make predictions about a patient’s health status. This information could then be used to further provide targeted and individualized interventions and improve health outcomes. In addition, machine learning mechanisms could be used in future studies to explain the high dropout rate of this study using available data on technology satisfaction, dropout reasons, and app usage behavior.

### 4.3. Conclusions

In conclusion, this study aimed to investigate the efficacy of individualized mHealth interventions in improving distress coping in healthcare professionals. The results showed positive effects on physical activity-related outcomes with high individualization, but no significant improvements on stress-related HRV parameters. The study highlights the importance of considering the context of physical activity and the need to differentiate between occupational and leisure-time physical activity. The results suggest that physical activity may be perceived as more stressful when it is part of work, rather than leisure time. The effectiveness of an app-based intervention seems to depend on design aspects and user-centeredness. Further research is needed to determine the optimal level of individualization and guidance for this type of intervention to effectively reduce stress levels.

## Figures and Tables

**Figure 1 sensors-23-02322-f001:**
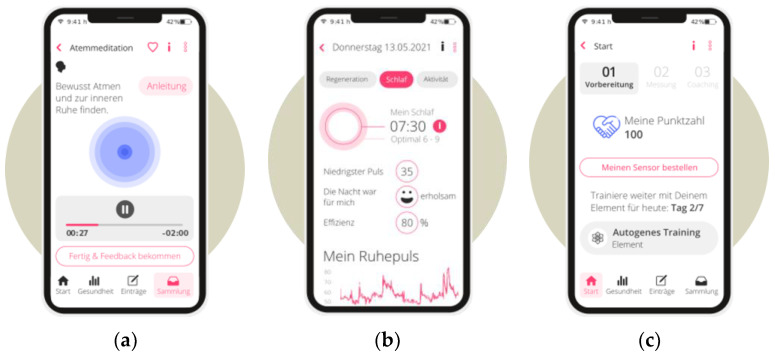
UI-Design of the App (**a**) breathing meditation user interface; (**b**) user interface of the landing page with a brief summary of vital parameters; (**c**) training progress user interface, Adapted with permission from fibase [98]. 2022, fitbase GmbH.

**Figure 2 sensors-23-02322-f002:**
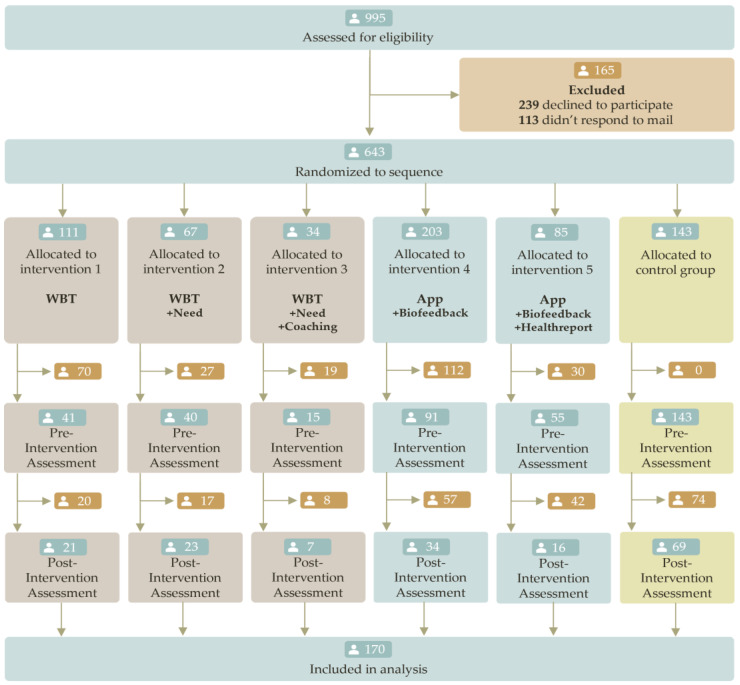
Consort study flow for multi-arm parallel group randomized controlled trials [92,93,94].

**Figure 3 sensors-23-02322-f003:**
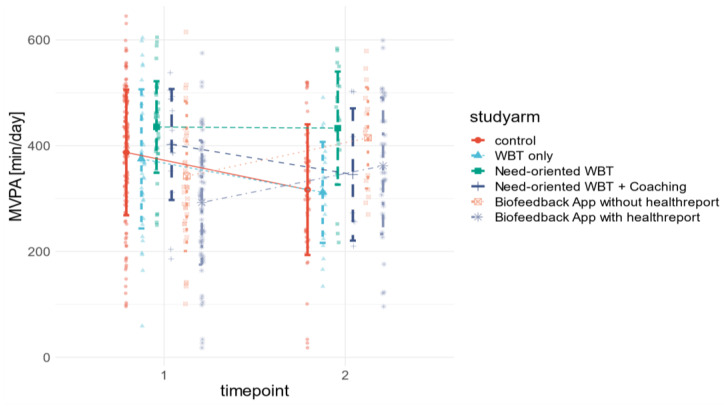
Grouped raincloud mean value plot of pre–post differences in moderate to vigorous physical activity [min/day] across study arms and control group.

**Figure 4 sensors-23-02322-f004:**
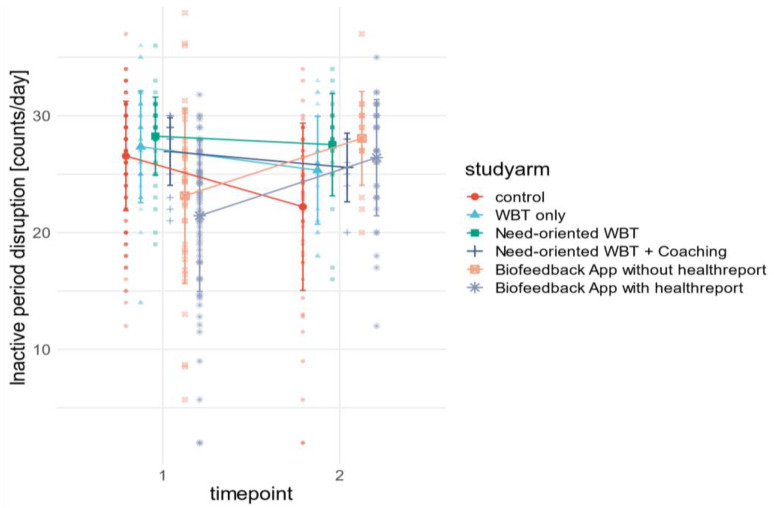
Grouped raincloud mean value plot of pre–post differences in inactive period disruption [counts/day] across study arms and control group.

**Table 1 sensors-23-02322-t001:** Brief description of intervention groups.

No.	Intervention	Type	Need	Biofeedback	Coaching	Report
1	Web-based digital stress management intervention	Web-based	No	No	No	No
2	Web-based need-oriented digital stress management intervention	Web-based	Yes	No	No	No
3	Web-based need-oriented digital stress management intervention with telephone coaching	Web-based	Yes	No	Yes	No
4	App-based personality specific digital stress management interventions with sensory biofeedback	App-based	No	Yes	No	No
5	App-based personality specific digital stress management intervention with sensory biofeedback and health report	App-based	No	Yes	No	Yes

**Table 2 sensors-23-02322-t002:** Detailed list of study arm specific intervention modules ((x) = it depends on study arm if this individualized feature occurs).

Focus	Sub Focus	App	WBT							
			Healthy Nutrition	Weight Loss	Physical Activity	Spine Gymnastics	Meditation/Mindfulness	Hatha Yoga	Sleep and Stress	Autogenic Training
**Individualization**	Direct biofeedback	x								
	AVEM patterns	x								
	Telephone coaching		(x)	(x)	(x)	(x)	(x)	(x)	(x)	(x)
	Health Report	(x)	(x)	(x)	(x)	(x)	(x)	(x)	(x)	(x)
	Need orientation		(x)	(x)	(x)	(x)	(x)	(x)	(x)	(x)
**Stress and** **relaxation**	Problem-focused	x								
	Deep breathing	x					x	x	x	x
	Mindfulness	x					x	x	x	
	Goal setting	x		x	x		x		x	x
	Gratitude journal	x					x	x	x	
	Positive psychology	x					x	x	x	x
	Autogenic training	x					x	x	x	x
	Muscle relaxation	x					x	x		
	Body perception	x					x	x		
	Stress physiology									x
**Physical** **activity**	Stretching and yoga	x			x			x		
	Fascia training			x	x	x				
	Behavior change	x								
	Activity habits	x								
	Endurance training	x			x					
	Anatomy			x	x	x				
	Spine health					x				

**Table 3 sensors-23-02322-t003:** Summary and description of relevant outcome parameters.

Parameter	Unit	Description [103]
SDNN	ms	Standard deviation of all RR intervals includes fluctuations over shorter as well as more widely divergent time periods.
RMSSD	ms	Square root of the squared mean value of the sum of all differences of successive RRintervals. Marker for selective assessment of efferent vagus activity and parasympathetic influence on the heart.
LF/HF ratio	%	Quotient of LF and HF: LF = power density spectrum from >0.04 to 0.15 Hz, percentage LF of the full spectrum. This parameter characterizes the potency of the low frequency components and can be attributed to parasympathetic as well as sympathetic activity; HF = power density spectrum from > 0.15 to 0.4 Hz, percentage HF of the full spectrum, mediated by respiratory-induced modulations of parasympathetic activity.
Baevsky	Index	Measure for characterizing recorded ECG signals or RR intervals. Reflects the degree of central control of the heart rhythm and characterizes the activity of the sympathetic part of the autonomic nervous system (VNS). It serves as an indicator of shifts in the balance of the VNS, i.e., changes in the balance between the effects of the sympathetic and parasympathetic nervous systems.
Steps	Counts/day	Accelerometer measured number of steps taken per day.
MVPA	Min/day	Accelerometer measured time spend in moderate to vigorous physical activity per day.
Disrupt	Counts/day	Accelerometer measured inactive period disruption counts. Counting occurs when a >30 min period of inactivity is interrupted with physical activity. This parameter serves as a measure of behavior change.
Inactivity	Min/day	Inactivity or sedentary behavior is defined by any waking behavior characterized by an energy expenditure ≤ 1.5 metabolic equivalents of task [METs] while in a sitting, reclining, or lying posture [104].

**Table 4 sensors-23-02322-t004:** Baseline values for Pre- and Post-Intervention Assessment and ANOVA statistics. Bold line values indicate significant time*group effects, which have been visualized in Figure 3 and Figure 4.

		Pre-Intervention Assessment							Post-Intervention Assessment							MANOVA		
		Studyarm					Control	Overall	Studyarm					Control	Overall	Time*Group		
		1	2	3	4	5			1	2	3	4	5			F(1,5)	*p*	η^2^p
Gender																		
male	n	12	4	2	43	21	20	102	7	0	1	3	3	52	66			
	%	29	10	13	47	38	14	26.5	33.3	0	14.3	18.7	8.8	75.4	38.8			
female	n	29	36	13	48	34	123	283	14	23	6	13	31	17	104			
	%	71	90	87	53	62	86	73.5	66.6	100	85.7	81.3	91.2	24.6	41.2			
Age	x¯	42.4	40.6	39.0	40.8	41.6	42.4	41.1	45.8	40.2	42.6	42.6	44.3	40.9	42.7	0.888	0.489	0.008
	s	12.1	11.2	9.8	10.6	11.5	10.2	10.9	10.6	9.5	9.3	9.7	10.7	10.5	10.0			
BMI	x¯	26.1	27.5	24.9	26.4	27.8	26.7	26.6	25.8	29.1	26.1	28.0	26.7	26.8	27.1	0.177	0.971	0.002
	s	7.1	5.3	4.6	6.6	7.0	6.0	6.1	3.9	9.0	5.0	7.4	6.3	6.1	6.3			
Steps counts/day	x¯	7925	8541	8535	7588	6609	8074	7879	6402	8720	8956	8129	6876	7562	7774	0.794	0.555	0.008
	s	4253	2827	3255	3025	2716	3509	3264	2745	3121	4774	3726	3010	3062	3406			
MVPA min/day	x¯	375.0	435.4	402.3	342.4	292.6	387.2	372.5	311.8	433.3	345.6	414.8	362.0	316.8	364.0	**5.826**	**<0.001**	**0.057**
	s	131.4	86.6	104.9	141.5	117.4	118.8	116.8	95.4	106.8	125.0	95.7	129.1	123.5	112.6			
Inactivity min/day	x¯	287.8	213.0	227.2	182.7	231.0	254.2	232.7	358.3	183.2	284.7	192.6	280.9	226.6	254.4	2.181	0.055	0.022
	s	150.3	113.5	138.5	81.1	112.4	137.5	122.2	156.1	108.3	137.0	72.1	142.7	120.2	122.7			
Disruption counts/day	x¯	27.3	28.3	26.9	23.2	21.4	26.6	25.6	25.3	27.5	25.6	28.1	26.4	22.2	25.9	**11.2**	**<0.001**	**0.100**
	s	4.8	3.3	2.9	7.5	6.5	4.7	4.9	4.6	4.4	2.9	4.0	5.0	7.2	4.7			
SDNN ms	x¯	50.3	47.3	47.3	49.5	49.7	48.9	48.8	50.6	47.0	43.6	47.2	48.0	51.2	47.9	0.609	0.693	0.006
	s	11.0	11.3	12.8	9.3	12.5	11.4	11.4	11.0	10.9	10.3	10.9	12.2	12.0	11.2			
RMSSD ms	x¯	28.4	28.5	27.4	28.3	29.3	27.9	28.3	28.6	27.3	27.0	26.1	27.6	29.9	27.7	0.697	0.626	0.007
	s	7.5	10.6	9.0	7.4	9.8	8.7	8.8	7.7	9.0	9.6	7.3	9.2	9.8	8.8			
LFHF %	x¯	5.1	4.9	5.1	5.7	4.8	5.0	5.1	4.6	5.2	4.4	6.1	4.7	4.9	5.0	0.214	0.956	0.002
	s	2.3	2.6	3.7	3.8	2.5	3.1	3.0	2.1	2.5	2.4	5.9	2.6	2.9	3.1			
Baevsky Index	x¯	241.3	279.0	283.1	268.0	270.5	263.9	267.6	225.4	289.3	307.4	258.6	282.4	248.5	268.6	0.196	0.964	0.002
	s	96.3	127.5	159.0	119.0	171.2	144.9	136.3	81.6	192.9	154.1	106.2	154.5	134.5	137.3			

## Data Availability

The datasets generated and analyzed during the study are available from the corresponding author on reasonable request.

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
