# Peer review of "Efficacy of Individualized Sensory-Based mHealth Interventions to Improve Distress Coping in Healthcare Professionals: A Multi-Arm Parallel-Group Randomized Controlled Trial"

_sensors, 2023, doi:10.3390/s23042322_

Round 1
Reviewer 1 Report
This paper studied the impact of mHealth interventions on distress coping in healthcare professionals. Several randomized controlled group trials were used in the study. The results revealed that stress related to heart rate variability did not change over time. The study concluded that the intervention duration was not enough in improving physical activity behavior. The study is interesting and in the scope of this journal. However, several changes/additions should be made such as:
- Please simplify the abstract. Omit redundant words/explanations. Don’t use uncommon abbreviations or just use the full words.
- Some statistics regarding stress levels or numbers can be presented at the beginning of the introduction section to show the importance of this research.
- Why improving physical activity behavior is important? How previous researchers used mHealth or other alternatives to cope with stress and its solution can be presented in the introduction.
- Summary of the main contributions of the study can be presented at the end of the introduction section. Using a point-by-point style would be beneficial for the readers.
- Section 2: Literature review/related studies with mHealth, health intervention, stress reduction, and physical behaviors can be presented there.
- Since this is a Sensors journal, the details presentation of the sensors used should be revealed as details as possible so that the readers can reproduce it easily.
- Most of the results are related to the participants of the study and there is no evaluation of the “Sensors” or “Devices” used in the study. Please provide those missing important results and discussions.
- Do you only use statistical methods? Why not use deep learning models to detect physical activities? Then evaluate the model and provide the performance evaluation results.
- The impacts of the developed model could also be much more interesting than only statistics.
Author Response
- Please simplify the abstract. Omit redundant words/explanations. Don’t use uncommon abbreviations or just use the full words.
Answer: Thank you for this essential suggestion. We shortened the abstract, simplified it, eliminated unusual abbreviations from it, and deleted redundant words and abbreviations. The revised version is now presented as follows: “Detrimental effects of chronic stress on healthcare professionals have been well-established, but the implementation and evaluation of effective interventions aimed at improving distress coping remains inadequate. Individualized mHealth interventions incorporating sensor feedback have been proposed as a promising approach. This study aimed to investigate the impact of individualized, sensor-based mHealth interventions focusing on stress and physical activity on distress coping in healthcare professionals. The study utilized a multi-arm, parallel group randomized controlled trial design, comparing five intervention groups (three variations of web-based training and two variations of an app training) that represented varying levels of individualization to a control group. Both self-reported questionnaire data (collected using Limesurvey) as well as electrocardiography and accelerometry-based sensory data (collected using Mesana Sensor) were assessed at baseline and post-intervention (after eight weeks). Of the 995 eligible participants, 170 (26%) completed the post-intervention measurement (Group 1: N=21; Group 2: N=23; Group 3: N=7; Group 4: N=34; Group 5: N=16; Control Group: N=69). MANOVA results indicated small to moderate time-by-group interaction effects for physical activity-related outcomes, including moderate to vigorous physical activity (F(1,5)=5.8, p=<.001, η²p=.057) and inactivity disruption (F(1,5)=11.2, p=<.001, η²p=.100), in the app-based intervention groups, but not for step counts and inactivity. No changes were observed in stress-related heart rate variability parameters over time. Despite a high dropout rate and a complex study design, the individualized interventions showed initial positive effects on physical activity. However, no significant changes in stress-related outcomes were observed, suggesting that the intervention duration was insufficient to induce physiological adaptations that would result in improved distress coping.“
- Some statistics regarding stress levels or numbers can be presented at the beginning of the introduction section to show the importance of this research.
Answer: Thank you for that comment. We have added information on the prevalence of stress levels in healthcare professionals to illustrate the necessity of this study: “In one study by Couarraze and colleagues, for instance, healthcare professionals indicated stress levels 25.8% higher than the general population.”(Page 1)
- Why improving physical activity behavior is important? How previous researchers used mHealth or other alternatives to cope with stress and its solution can be presented in the introduction.
Answer: Thank you for this thought-provoking comment. Physical activity is important in the context of distress coping because it alters the release of stress hormones, promotes relaxation and improves mental health (see page 3, line 98).We added a series of alternative mechanisms for distress coping to the introduction to address your concern: “Beside physical activity, many other ways to cope with stress are evident: Breathing exercises like diaphragmatic breathing [47], progressive muscle relaxation [48], Meditation [49], Yoga [50], gratitude journaling [51], Listening to music [52] and autogenic training [53].“
- Summary of the main contributions of the study can be presented at the end of the introduction section. Using a point-by-point style would be beneficial for the readers.
Answer: We are grateful for this reasonable complementary suggestion. To address this, we have added the following at the end of the introduction: " To the best of our knowledge, this study complements the described existing body of research and is the first to:
- ...scientifically validate a sensor-based mHealth intervention for distress coping in the healthcare setting.
- ...compare the efficacy of different individualization levels of mHealth interventions by a multi-arm study design.
- ... combine components of physical activity and relaxation techniques in an mHealth application using a multimodal intervention approach to improve distress coping among healthcare professionals.
- ... measure multiple clinically relevant stress and physical activity-related outcomes during the intervention and adapt the content of the intervention based on these measures.
- ... enable unrestricted implementation within the daily work routine of healthcare professionals by means of a mobile and low-threshold intervention.”
- Section 2: Literature review/related studies with mHealth, health intervention, stress reduction, and physical behaviors can be presented there.
Answer: Even though it is unclear what is meant by "section 2", we tried to address your point about too few references and added the following paragraph in addition to the existing references on mHealth effectivity: “Beside physical activity, many other ways to cope with stress are evident: Breathing exercises like diaphragmatic breathing [48], progressive muscle relaxation [49], Meditation [50], Yoga [51], gratitude journaling [52], Listening to music [53] and autogenic training [54].“
- Since this is a Sensors journal, the details presentation of the sensors used should be revealed as details as possible so that the readers can reproduce it easily.
Answer: The sensor information described in the methodology section (page 9, line 387-390) is regrettably as much information as we are allowed to provide for patent infringement reasons. Appropriate sources are given in this section to validate the sensors: : The assessment applied a selection of standardized questionnaire measures as well as sensor-based physiological and vital parameter measures (measured by Corvolution CM300 [99], which includes ECG circuit, 3-axis acceleration and rotation rate chip, air pressure chip, thoracic impedance chip and temperature chip) [100,101]. The sensitivity of patch-based ECG sensors such as this is 93.4-97.0%, and the specificity is 95.6-98.8% [89].
- Most of the results are related to the participants of the study and there is no evaluation of the “Sensors” or “Devices” used in the study. Please provide those missing important results and discussions.
Answer: We share this view and will report on a number of user experiences in additional research papers.. However, the focus here is on the effectiveness of the intervention. A validation of the sensor has already been done (see reference 99 in the Methods section). In addition we surveyed the app usage behaviour, dropout reasons and satisfaction with the technical intervention (CSQI questionnaire) in this study. As the dropout analysis is very extensive due to the complexity of the trial, it will be the subject of further publications on this study. We additionally highlighted this in the “future research section”: "In addition, machine learning mechanisms could be used in future studies to explain the high dropout rate of this study using available data on technology satisfaction, dropout reasons and app usage behaviour.”
- Do you only use statistical methods? Why not use deep learning models to detect physical activities? Then evaluate the model and provide the performance evaluation results. The impacts of the developed model could also be much more interesting than only statistics.
Answer: We appreciate this suggestion! The use of machine learning would certainly be useful for further individualization and should be addressed in future studies. However, the study design of the present study aims at the use of statistical methods. We have added the following paragraph in the section "Future research": “Furthermore, machine learning elements could improve future mHealth interventions by providing personalized and data-driven solutions for health problems. Machine learning algorithms can analyze large amounts of data from various sources, such as electronic health records, wearable devices, and mobile apps, to identify patterns and make predictions about a patient's health status. This information could then be used to further provide targeted and individualized interventions and feedback of the progress and in turn improve health outcomes.“
Reviewer 2 Report
I am very excited to review the manuscript and want to thank the authors for their interesting work. Though the authors did great work in the current version of this manuscript, I think the following comments will helpful for them to improve better for this journal:
1. In the abstract section, they mentioned, "Despite high dropout rates and complex study 25 design, individualized interventions revealed initial effects on physical activity, but not the expected 26 effects on stress-related outcomes." I recommend adding some description in the discussion or another section on how it can be resolved.
2. In the introduction section, they mentioned that, "In general, psychosocial stressful stimuli activate neuronal, neuroendocrine, and endocrine pathways." So mHealth technology can be helpful for improving the psychosocial stress. The following book is a good example of this:
"Rabbani, Masud, et al. "A Mobile Health Application for Monitoring Children With Autism Spectrum Disorder: ASD Monitoring by mHealth." AI Applications for Disease Diagnosis and Treatment. IGI Global, 2022. 40-65."
3. In the introduction section, they mentioned that, "For instance, mindfulness programs improve quality of life, anxiety, stress perception and sleep quality [39,40]". Yes the mHealth tool can be useful for improving the quality of the life
4. In the introduction section, they mentioned that, "However, there are also hybrid forms such as web apps." But there need some reference which are missing in this section. Here are some good examples:
"Rabbani, Masud, et al. "A data-driven validation of mobile-based care (mCARE) project for children with ASD in LMICs." Smart Health 26 (2022): 100345."
Authors can cite these works as a reference for the hybrid forms of intervention.
5. The last paragraph of the introduction section should be reformed. Authors can add their contribution to this work by pointwise in this paragraph.
6. In the participants section, authors should add a demographic table on the participant and should describe on the IBR information and consent information from the patient.
7. Is "Figure 1 " created by the authors by themselves; otherwise, they need to redraw the figure by themselves.
8. I think the authors need to add a conclusion section.
Author Response
- In the abstract section, they mentioned, "Despite high dropout rates and complex study 25 design, individualized interventions revealed initial effects on physical activity, but not the expected 26 effects on stress-related outcomes." I recommend adding some description in the discussion or another section on how it can be resolved.
Answer: Thank you very much for this remark. To address your concern, we have included the following suggestion regarding the design of future mHealth trials with high dropout rates in the "Future research" paragraph: “To reduce the dropout rate, future studies could offer incentives, simplify the study process, improve face-to-face communication, monitor and address adverse events and foster a more positive research culture.”
- In the introduction section, they mentioned that "In general, psychosocial stressful stimuli activate neuronal, neuroendocrine, and endocrine pathways." So mHealth technology can be helpful for improving the psychosocial stress. The following book is a good example of this: "Rabbani, Masud, et al. "A Mobile Health Application for Monitoring Children With Autism Spectrum Disorder: ASD Monitoring by mHealth." AI Applications for Disease Diagnosis and Treatment. IGI Global, 2022. 40-65."
Answer: Thank you, we cited the suggested reference accordingly.
- In the introduction section, they mentioned that "For instance, mindfulness programs improve quality of life, anxiety, stress perception and sleep quality [39,40]". Yes the mHealth tool can be useful for improving the quality of the life
Answer: We are not quite sure how to understand your comment. We have written a whole paragraph elsewhere in the introduction (p.4, line 174) on the effectiveness of mHealth: “mHealth Interventions can be a low-threshold opportunity for health promotion and are a promising possibility to achieve prevention goals [74]. The free allocation of time and flexibility of availability were evaluated on a positive note. Combining such apps with so-called “wearables”, such as smartwatches or fitness trackers, could be a promising approach to continuously record health data and thus constitute various opportunities in the context of prevention work (Gamification, Just in time adaptive interventions). By implementing wearable devices into mHealth applications (apps), health-related data (e.g., sleep patterns, eating patterns, and exercise) could be recorded and respective need-tailored interventions be derived [75]. Previous studies already showed positive effects of stress apps on wellbeing. Harrer et al. [76] for example found that app-based stress management interventions improved stress, anxiety, and depression in college students. Another example stems from research by Economides et al. [77] who found that a mindfulness app intervention reduced stress and irritability, while it also increased positive affect. At the same time, expectations towards health apps are high, 70% of health app users believe that these can strengthen self-motivation and 56% think that app use can improve health education [78].“
- In the introduction section, they mentioned that "However, there are also hybrid forms such as web apps." But there need some reference which are missing in this section. Here are some good examples: "Rabbani, Masud, et al. "A data-driven validation of mobile-based care (mCARE) project for children with ASD in LMICs." Smart Health 26 (2022): 100345." Authors can cite these works as a reference for the hybrid forms of intervention.
Answer: Thank you for this stimulating piece of literature. We have included it in the introduction as follows and thereby hope to have addressed your feedback: One example for a hybrid approach could be the mCARE project by Rabbani et al. [72]. The project discusses a data-driven validation of a mobile-based care project (mCARE) aimed at helping children with Autism Spectrum Disorder (ASD) in Low and Middle-Income Countries (LMICs). The results showed that the mCARE project had a positive impact on the children’s symptoms and behavior and was effective in reducing the burden on caregivers. The study provides evidence for the potential of mobile technology to improve access to care for children with ASD in LMICs.
- The last paragraph of the introduction section should be reformed. Authors can add their contribution to this work by pointwise in this paragraph.
Answer: We are grateful for this reasonable complementary suggestion. To address this, we have added the following at the end of the introduction: " To the best of our knowledge, this study complements the described existing body of research and is the first to:
- ...scientifically validate a sensor-based mHealth intervention for distress coping in the healthcare setting.
- ...compare the efficacy of different individualization levels of mHealth interventions by a multi-arm study design.
- ... combine components of physical activity and relaxation techniques in an mHealth application using a multimodal intervention approach to improve distress coping among healthcare professionals.
- ... measure multiple clinically relevant stress and physical activity-related outcomes during the intervention and adapt the content of the intervention based on these measures.
- ... enable unrestricted implementation within the daily work routine of healthcare professionals by means of a mobile and low-threshold intervention.”
- In the participants section, authors should add a demographic table on the participant and should describe on the IBR information and consent information from the patient.
Answer: The baseline demographic information is outlined in Table 4 of the results section according to CONSORT guidelines, while details on consent and IBR are included at the end of the document as per the sensor template's guidelines.
- Is "Figure 1 " created by the authors by themselves; otherwise, they need to redraw the figure by themselves.
Answer: Thank you für this note. The images were extracted from the app used with the explicit permission from the cooperating partner Fitbase. We have added the developer information and cited the app as software.
- I think the authors need to add a conclusion section.
Answer: Thank you very much for this essential advice. We added a conclusion paragraph at the end of our manuscript: “In conclusion, this study aimed to investigate the efficacy of individualized mHealth interventions in improving distress coping in healthcare professionals. The results showed positive effects on physical activity-related outcomes with high individualization, but no significant improvements on stress-related HRV parameters. The study highlights the importance of considering the context of physical activity and the need to differentiate between occupational and leisure-time physical activity. The results suggest that physical activity may be perceived as more stressful when it is part of work, rather than leisure time. The effectiveness of an app-based intervention seems to depend on design aspects and user-centeredness. Further research is needed to determine the optimal level of individualization and guidance for this type of intervention to effectively reduce stress levels.”
Round 2
Reviewer 1 Report
Well addressed. I think so.